# Cutaneous Deep Ulcerations as Initial Presentations of Granulomatosis with Polyangiitis: Two Case Reports and Differential Diagnosis

**DOI:** 10.3390/medicina59030563

**Published:** 2023-03-14

**Authors:** Jiandan Qian, Jiawen Li, Jun Li, Guiqiang Wang, Hong Zhao

**Affiliations:** 1Department of Infectious Diseases and the Center for Liver Diseases, Peking University First Hospital, Beijing 100034, China; emerallqian@163.com (J.Q.);; 2Department of Infectious Diseases and the Center for Liver Diseases, Peking University International Hospital, Beijing 102206, China; 3The Collaborative Innovation Center for Diagnosis and Treatment of Infectious Diseases, Zhejiang University, Hangzhou 310003, China

**Keywords:** granulomatosis with polyangiitis, skin, ulcer, differential, diagnosis

## Abstract

*Background*: Granulomatosis with polyangiitis (GPA) is an antineutrophil-cytoplasmic-antibody (ANCA)-associated small-vessel vasculitis characterized by necrotizing granulomatous inflammation. Symptoms of skin involvement can appear in 30–50% of patients with GPA, and may present as the initial presentation. *Case Presentation*: We describe two patients who presented with multiple deep, large, nonhealing skin ulcers postoperatively with purulent drainage and fever. Both patients were diagnosed with GPA after an extensive evaluation, including histopathology. Infectious, connective tissue disease and malignant etiologies were excluded. Their cANCA and PR3-ANCA levels were positive. Patient 2 was diagnosed early and recovered well after treatment with corticosteroids and rituximab; however, Patient 1 had a poor prognosis due to a long disease course. *Conclusions*: Diseases with multiple deep, large skin ulcers and fever can be infectious or noninfectious. Atypical manifestations may lead to missed diagnosis and misdiagnosis. GPA may initially present in a localized form before progressing to a generalized disease. The two cases we have highlighted will prompt clinicians to nevertheless call for a low threshold for diagnosis.

## 1. Introduction

Granulomatosis with polyangiitis (GPA), formerly known as Wegener’s granulomatosis (WG), is a systemic necrotizing vasculitis involving small and medium blood vessels throughout the body [1]. Antineutrophil cytoplasmic antibodies (ANCAs) have a substantial connection with GPA, and the cytoplasmic labeling pattern (cANCA), which is directed against proteinase 3 (PR3), has a positive rate of 90% in patients with generalized disorders. GPA is a relatively rare disease with an estimated prevalence of 200–400 cases per one million people [2]. The incidence and prevalence of GPA varies by region and ethnicity, with a higher incidence in northern Europe and a lower incidence in Asian countries [3]. The prevalence of GPA in China is 1.94/100,000 in the population and is similar between males and females. The mean age at onset of GPA in China is 40 years, which is similar to published data from other countries [4]. Although peak incidence occurs at middle age, pediatric GPA has also been reported [5]. All body organ systems can be involved in GPA, mainly the upper and lower respiratory tract and kidneys. Rapidly progressive glomerulonephritis and hemorrhagic alveolitis are the leading causes of morbidity and mortality in GPA [6]. Skin involvement symptoms occur in 30–50% of individuals and may present as the initial presentation [7]. Cutaneous vasculitis secondary to GPA is polymorphic, mostly presenting as palpable purpura (especially in the lower limbs), followed by subcutaneous nodules, papules, blisters, vesicles, necrotic-ulcerative lesions, livedo reticularis, and even pyoderma gangrenosum (PG)-like ulcers [8,9,10,11]. Multiple types of GPA skin manifestations can coexist in the same patient, and different skin lesions may appear in different stages of GPA [8].

Here, we report two cases of recurrent purulence at the wound site after surgery, accompanied by deep, large skin ulcers at multiple sites, gradually developing into multi-organ involvement. A skin biopsy suggested granulomatous vasculitis. GPA was confirmed when combined with positive c-ANCA and PR3-ANCA levels.

## 2. Case Presentation

### 2.1. Case 1

A 59-year-old woman presented to the infectious disease clinic with recurrent ulceration and discharge of pus from a surgical incision in the gallbladder and uterus made three years ago, with intermittent fever and abdominal pain and a maximum temperature of 39.5 °C. Drainage of the abscess temporarily relieved her symptoms. Bacterial, fungal, and acid-fast bacilli cultures of the drainage fluid were all negative. Antibiotic treatment was ineffective. The patient’s symptoms began to worsen one month before admission, with multiple large nonhealing ulcers on her abdominal wall and scapula. The patient gradually developed pulmonary manifestations, such as cough, sputum, and wheezing, as well as hearing impairment in both ears. She had undergone a hysterectomy and double adnexectomy for uterine fibroids 9 years ago and a cholecystectomy 5 years ago.

She still had fever after admission, with a temperature of 38.1 °C. Multiple ulcerative lesions with purulent viscous secretions were seen on the abdominal wall (Figure 1). Laboratory examinations found that CRP, ESR, and other inflammatory indicators were significantly increased, and urinalysis and serum creatinine were normal. ANA, anti-extractable nuclear antigen (ENA) profiles, complement, rheumatoid factor (RF), and immunofixation electrophoresis were all within the normal range or negative. Repeated blood culture, sputum culture, sputum for tuberculosis, and sputum for fungus were negative. Pure-tone audiometry indicated binaural mixed hearing loss. Abdominal and pelvic CT scans showed multiple skin defects and abdominal subcutaneous abscesses (Figure 1). A chest CT scan revealed multiple pulmonary nodules and masses, and thick-walled cavities were formed (Figure 1).

The patient underwent a bronchoscopy procedure with a bronchoalveolar lavage (BAL), with negative results for Gram staining, cultures, and acid-fast bacilli staining, as well as absent growth on fungal and mycobacterial cultures. Bronchoscopy biopsy revealed small-blood-vessel vasculitis, with neutrophil infiltration. Histology of a skin specimen showed perivasculitis and granulomatous vasculitis (Figure 2). Immunofluorescence examination revealed c-ANCA positivity (1:10), PR3-ANCA > 200 IU/mL, and granulomatosis with polyangiitis (GPA) was diagnosed.

The patient was temporarily in remission under corticosteroid and cyclophosphamide (CYC) treatment, with normal temperature and gradually healed skin lesions. Unfortunately, she relapsed with headache. A cranial MRI found multiple cavity infarctions, and a lumbar puncture showed that cerebrospinal fluid pressure was 220 mm H_2_O. The dose of steroids was increased to 60 mg/d, 20 g of gamma globulin was intermittently infused, and 100 mg/day azathioprine was added for remission. The PR3-ANCA titer decreased to a minimum of 79 RU/mL and symptoms were relieved after adjusted treatment. However, the headache and earache continued to worsen intermittently, and she suffered from aggravation of lung disease and three intestinal obstructions during the 9-month follow-up. The intestinal obstructions were believed to be caused by CYC drug intolerance and improved after CYC withdrawal; details are shown in Figure 1.

### 2.2. Case 2

A 62-year-old woman presented to us because of a five-month history of generalized nodular abscess and intermittent fever that developed one month after hysterectomy surgery, and the abscess burst one month before admission. Persistent pink liquid discharge from the vagina occurred after surgery, and the patient began to develop a painless skin mass on the right upper arm approximately 2 cm in diameter one month later. Almost ten similar masses gradually appeared on other parts of her body. Four months later, the skin lesion gradually turned red and purple and then broke, purulence flowed out in the form of a rice-soup-like liquid, and the ulcerative lesions became larger and deeper (Figure 3). Oliguria and pitting edema of both lower extremities developed one week before admission. A biopsy of the skin was performed in the outpatient department and showed granulomatous necrotizing vasculitis with thrombosis and mixed inflammatory cell infiltration, and acid-fast staining was negative.

Physical examination showed multiple deep ulcers throughout the body (Figure 3). ESR, CRP, and other inflammatory indicators were significantly increased. Urinary sediment results showed microalbuminuria, WBC 60–70/HP, and deformable RBCs 8–10/HP. The maximum value of serum creatinine was 120 μmol/L(normal range 44–133 μmol/L), and the eGFR was 41.83 mL/min/1.73 m^2^. Pus secretion PCR performed by the laboratory of the outpatient department found *Mycobacterium tuberculosis* (MTB), *Mycobacterium avium*, and *Mycobacterium ulcerans* with low titers. However, pus and blood NGS and cultures of fungi, bacteria, nocardia, and actinomyces were all negative. Urine, pus, and blood Roche cultures of MTB and non-Mycobacterium tuberculosis (NTM) were negative after 60 days of extension. Abdominal and pelvic CT showed multiple abdominal skin defects, subcutaneous abscesses, and fluid hypodense areas in the vaginal stump (Figure 3). A chest CT showed mild interstitial changes in the right lung. A brain MRI revealed bilateral ethmoid and maxillary sinus inflammation. After experimental treatment with quadruple anti-tuberculosis drugs, the symptoms did not improve, and the inflammatory indices and serum creatinine increased progressively. An immunofluorescence examination revealed c-ANCA positivity (1:10) and PR3-ANCA (ELISA) > 200 IU/mL. Repeated histology of a skin specimen was consistent with GPA (Figure 2), vaginal biopsy showed basophilic granuloma, and renal puncture was consistent with ANCA-associated polyangiitis renal impairment. A diagnosis of GPA was suggested.

Treatment was combined corticosteroid with rituximab. Her body temperature dropped to normal during follow-up, and the ulcers gradually healed (Figure 4). Serum creatinine and eGFR were stable and within the normal range. The patient’s treatment response and prognosis during hospitalization are shown in Figure 3.

## 3. Discussion

Patient 1 (hearing loss +1, ANCA or PR3-ANCA positivity +5, characteristic chest imaging +2, granuloma on biopsy +2; total score 10) and Patient 2 (ANCA or PR3-ANCA positivity +5, granuloma on biopsy +2, inflammation of the nasal sinuses on imaging +1, pauci-immune glomerulonephritis +1; total score 9) described in this article initially presented with multiple skin ulcerative lesions and met the 2022 American College of Rheumatology/European Alliance of Associations for Rheumatology classification criteria for GPA [12]. After comprehensive and systematic examination excluded infection, both patients were diagnosed with GPA. Patient 2 was diagnosed early and recovered well after treatment with corticosteroid and rituximab; however, Patient 1 had a poor prognosis due to a long disease course.

Cutaneous manifestations are common and diverse in AAV and correlate with disease severity in patients with GPA. Severe systemic manifestations of vasculitis, such as alveolar hemorrhage and glomerulonephritis, are more common in GPA patients with skin lesions than in patients without skin manifestations [7]. Studies have shown that ulcerative lesions are a relatively rare presentation in GPA cutaneous involvement, ranging from 9.5% to 35.3% [13,14,15,16,17], and deep and large skin ulcers are rare in GPA patients, which can easily lead to missed diagnosis and misdiagnosis. The two patients described in this article highlight the importance of clinicians being knowledgeable of the clinical features and differential diagnoses of deep, large ulcerative skin lesions, fever, and multiorgan dysfunction. It also emphasizes the need to be familiar with skin manifestations from systemic diseases.

The differential diagnosis for refractory cases of cutaneous GPA must include indolent infectious etiologies, such as *Mycobacterial, Nocardia, Actinomyces, Leishmania donovani*, and deep fungal infection. A microbiological analysis is necessary whenever an infection is suspected. Due to the fact that *Nocardia, Actinomyces, Leishmania donovani*, and deep fungus were not found in either patient using repeated multiple cultures or NGS, the possibility of infection of these pathogens was considered to be low. This paper mainly discusses the differential diagnosis with *Mycobacterium*.

MTB infection has a high prevalence in our country, with more than 100,000 new cases in 2016 [18]. Small-vessel vasculitis secondary to MTB should be a differential diagnosis [19]. Of all patients who present with extrapulmonary manifestations of tuberculosis, 1% to 2% suffer from cutaneous tuberculosis (CTB) [20]. *Mycobacterium* tuberculosis cutaneous lesions are nonpathognomonic, but deep ulceration is rare. A study showed that BCG vaccination or immunotherapy may be correlated with local cutaneous complications, such as nonhealing ulcers and localized abscesses [21]. Additionally, visceral tuberculosis is rarely associated concurrently with cutaneous involvement [22]. Biopsies of cutaneous lesions for the identification of acid-fast staining bacilli and cultures represent the cornerstone of diagnosis and differential diagnosis.

However, skin is a common site of NTM infection in both immunocompromised and immunocompetent individuals [23]. Cutaneous NTM infection presents with localized lesions on the extremities in immunocompetent patients, whereas it presents with disseminated cutaneous lesions in immunocompromised patients [24]. Skin lesions can manifest as cellulitis, papular lesions, nodules, abscesses, draining sinuses, and ulcerations [25]. Diagnosis relies on a positive tissue culture for NTM and histopathological features, which have often been described as nonspecific granulomatous dermatitis, dermal granulomatous inflammation accompanied by necrosis and suppuration, and suppurative inflammation with little granuloma formation and numerous acid-fast bacilli [24].

In this article, pus secretion PCR for Patient 2 found three types of mycobacteria. Considering that a patient is less likely to be infected with several mycobacteria at the same time, multiple-sample Roche cultures of MTB and NTM were negative after 60 days of extension. A low positive titer of Mycobacterium does not exclude the possibility of contamination. Combined with effective treatment with corticosteroids and immunosuppressants, a diagnosis of noninfectious vasculitis was suggested.

In terms of noninfectious diseases, ulcerative PG should also be a differential diagnosis. Ulcerative lesions mimicking PG have been reported as a rarer cutaneous presentation of systemic GPA, and PG-like ulcers may present as the initial presentation or evolve from pustules, papulo-necrotic lesions, nodules, blisters, erythematous macules, or “swelling” progress [10]. The epidemiological data on the actual incidence rate of PG-like ulcers in GPA are controversial. In a retrospective analysis of 743 patients with GPA, 1.1% of cases had skin ulcerations resembling PG [26]. Another study reviewed 244 patients with GPA; among them, 30 patients had cutaneous GPA, and 26.7% (8/30) had PG-like ulcers, constituting the second most common skin manifestation after palpable purpura [14]. Shakshouk H et al. [11] retrospectively analyzed 211 patients who had cutaneous manifestations attributable to AAV, and 4.3% (9/211) of patients were identified to have PG-like ulceration based on specific skin lesions. Among those, six patients (66.7%, 6/9) were diagnosed with GPA. GPA was more frequently associated with PG-like ulcers than other AAVs.

Some clues may be useful for differentiation. First, the most common ulcerative variant of PG typically presents with single or multiple painful ulcerations with irregularly raised, undermined erythematous–violaceous borders [10,11,27]. Second, the visceral involvement of PG is relatively rare. Due to the subsequent typical visceral involvement, some patients initially diagnosed with PG were later reclassified as GPA [28]. Third, histopathological features can also provide helpful clues. The presence of a dense neutrophilic infiltrate without vasculitis is typical for PG [10], but leukocytoclastic vasculitis is the most consistent feature of skin lesions in GPA [11].

Cutaneous intravascular NK/T-cell lymphoma can cause painful erythema and nodules on the skin and should also be differentiated from GPA [29,30]. However, the absence of clonal histopathological findings and molecular biology in GPA excludes the possibility of neoplastic disease.

## 4. Conclusions

Diseases with multiple deep, large skin ulcers without other typical manifestations are indeed a diagnostic challenge to the physician. Both infectious and noninfectious factors need to be considered. Although many classification criteria have been developed to help with the prompt diagnosis of GPA, there are also instances in which these diseases may present in atypical ways. GPA may initially present in a localized form before progressing to a generalized disease; the two cases we have highlighted had multiple deep, large cutaneous ulcers as the initial presentation. Hopefully, this will prompt clinicians to nevertheless call for a low threshold for diagnosis, so that the disease can be recognized early and treated effectively to avoid missed diagnosis and misdiagnosis.

## Figures and Tables

**Figure 1 medicina-59-00563-f001:**
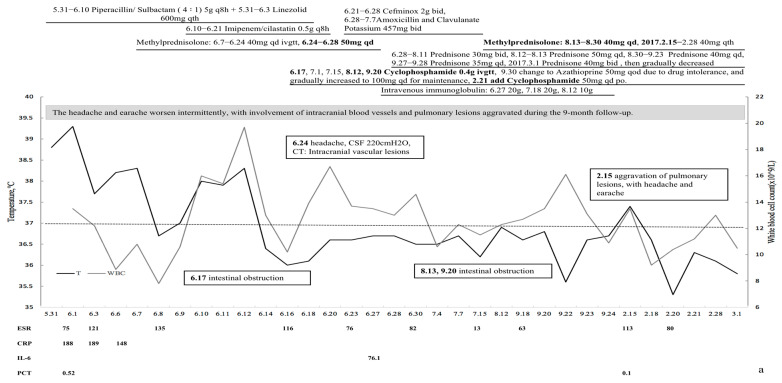
Clinical manifestations and computed tomography (CT) imaging of Patient 1. (**a**) Treatment response and prognosis of the patient. (**b**,**c**) Ulcerative lesions on the abdominal wall. (**d**–**g**) Abdominal and pelvic CT scans. Skin defect on the abdomen (white arrow), with multiple abdominal subcutaneous abscesses (black arrows). (**h**–**k**) Chest CT scan. Pulmonary nodules (white arrows), with thick-walled cavities (black arrows).

**Figure 2 medicina-59-00563-f002:**
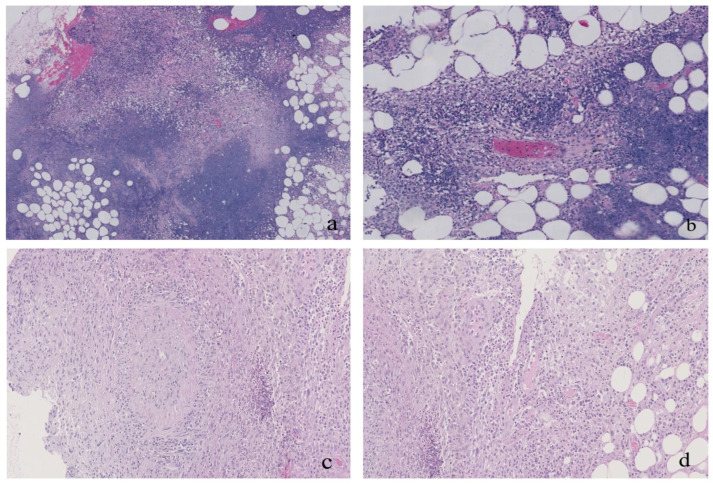
Histology of skin specimens (H&E stain) of Patient 1 (**a**,**b**) and Patient 2 (**c**,**d**). (**a**) Many lymphocytes and neutrophils infiltrated the dermis and subcutaneous fat layer, multinucleated giant cells aggregated, and small abscesses and fibrous fat map-like necrosis were observed (original magnification ×4). (**b**) Small vasculature was obliterated, and endothelial hyperplasia, neutrophil infiltration in the wall, Langham-like multinuclear giant cell aggregation, perivasculitis, vasculitis, and granulomatous vasculitis can be seen in the deep dermis and near the subcutaneous necrosis area (original magnification ×10). (**c**) Diffuse mixed-inflammatory-cell infiltration can be seen in the deep dermis and subcutaneous fat. Most of the histiocytes, plasma cells, neutrophils, and lymphocytes can be seen (original magnification ×110). (**d**) A medium-sized blood vessel in subcutaneous fat was obliterated, and there was histiocyte infiltration and granuloma formation on the vessel wall. Thrombosis and neutrophil infiltration can be seen in some small vascular lumens (original magnification ×110).

**Figure 3 medicina-59-00563-f003:**
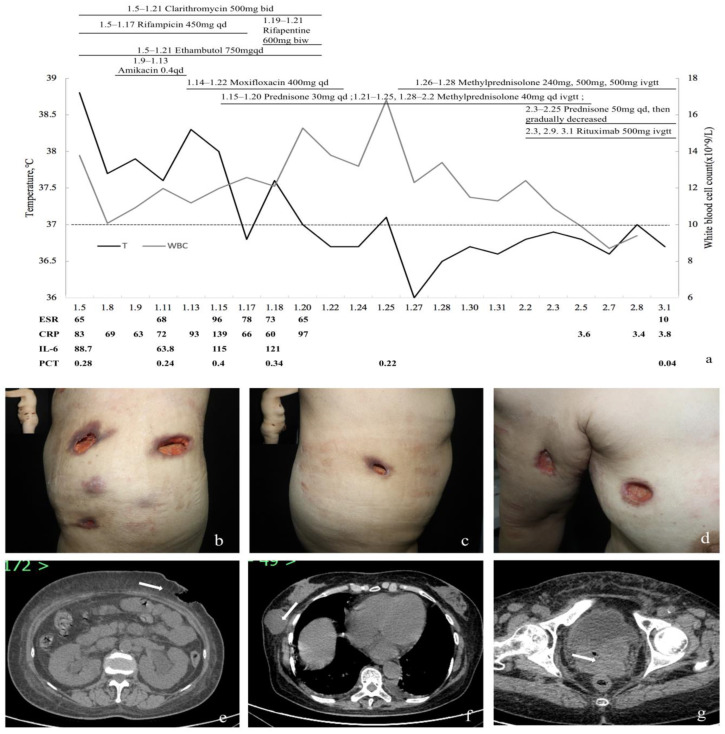
Clinical manifestations and computed tomography (CT) imaging of Patient 2. (**a**) Treatment response and prognosis of the patient. (**b**–**d**) Ulcerative lesions on the left loin (**b**), right loin (**c**), and right upper arm and chest (**d**). (**e**–**g**) CT examination of the abdominal and pelvic cavities. The arrow in e indicates abdominal wall skin defects, the arrow in f indicates subcutaneous masses, and the arrow in g indicates pelvic masses.

**Figure 4 medicina-59-00563-f004:**
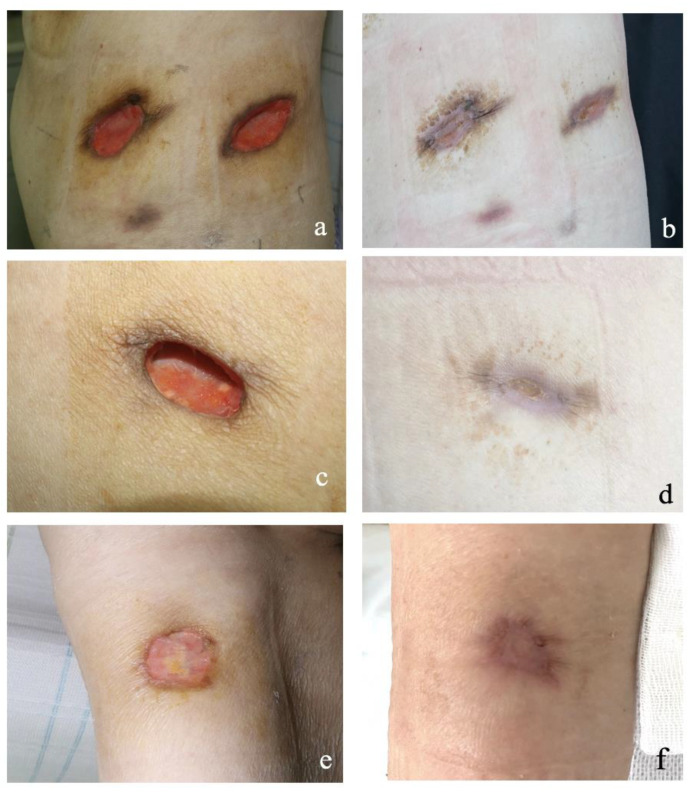
Changes in skin lesions during follow-up in Patient 2 (**a**–**f**). Skin lesions of the left loin (**a**) 26 January 2021, (**b**) 24 February 2021, right loin (**c**) 26 January 2021, (**d**) 3 March 2021, and right upper arm (**e**) 26 January 2021, (**f**) 1 March 2021.

## Data Availability

Data sharing is not applicable.

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
