# Peer review of "Cutaneous Deep Ulcerations as Initial Presentations of Granulomatosis with Polyangiitis: Two Case Reports and Differential Diagnosis"

_medicina, 2023, doi:10.3390/medicina59030563_

Round 1
Reviewer 1 Report
1. The words in Figure 1a is difficult to see. Please correct this.
2. Please provide the chest CT picture and Chest X-ray figure of the case 1.
3.In Figure 2, which figures belong to case 1 or case 2? It needs clearly marked.
4. In line 106, the description of the course of disease in case 1 is not complete.
5. Any information for possible renal involvement (urinary sediment result or proteinuria?) in case 1 and 2.
6. The words in Figure 2a is difficult to see. Please correct this.
7. In case 2, the information about renal function should be provided.
8. Please change the supplementary figure to Figure 4.
9. A brief discussion about previous case report Ref 5-7 and the articles Clin Exp Dermatol. 2022 Sep;47(9):1716-1719. doi: 10.1111/ced.15251 is needed.
10. The conclusion needs revised. It should be more informative for the doctors.
11. In the introduction, the epidemiological data in China could be provided.
Reviewer 2 Report
I read this case reports with interest. Here are my comments.
1. Did you submit it for English proofreading? Many small mistakes can be seen.
2. PR3 should be revised to PR3-ANCA.
3. Please indicate the values for CRP and ESR in both cases.
4. The clinical course text is too small to read.
5. Which is Case 1 and which is Case 2 in Figure 2?
Round 2
Reviewer 1 Report
The authors have response to my comments. It is a nice paper.
Reviewer 2 Report
The authors addressed to the questions raised by this reviewer.